# Artificial intelligence applications in refractive error management: A systematic review and meta-analysis

Josephine Ampong[1], Sylvia Agyekum[1], Werner Eisenbarth[2],
Albert Kwadjo Amoah Andoh[1¤a], Isaiah Osei Duah Junior[1¤b], Gabriel Amankwah[3¤c],
Gabriel Kwaku Agbeshie[1], Eldrick Adu Acquah[1¤d], Clement Afari[1], Emmanuel Assan[1],
Saphiel Osei Poku[1], Karen Ama Sam[1], Josephine Ampomah Boateng[1],
Kwadwo Owusu Akuffo[1]*

**1** Department of Optometry and Visual Science, College of Science, Kwame Nkrumah University of Science and Technology, Kumasi, Ghana, **2** Department of Applied Sciences and Mechatronics, Hochschule München University of Applied Sciences, Munich, Germany, **3** Department of Chemistry, College of Science, Kwame Nkrumah University of Science and Technology, Kumasi, Ghana

¤a Galaxy Eye Care Limited, Sekondi-Takoradi, Ghana,
¤b Department of Biological Sciences, College of Science, Purdue University, West-Lafayette, Indiana, United States of America,
¤c Department of Chemistry, College of Science, Purdue University, West-Lafayette, Indiana, United States of America,
¤d College of Optometry, University of Houston, Houston, Texas, United States of America
* akuffokwadwoowusu@knust.edu.gh, koakuffo@gmail.com

## Abstract

Artificial intelligence (AI) has transformed healthcare, and is becoming increasingly useful in eye care. We conducted a systematic review and meta-analysis of the use of AI in the diagnosis, detection, prediction, progression, and treatment of refractive errors (REs). The study adhered to the PRISMA checklist to ensure transparent reporting. The following databases were searched from inception to January 2025, with an English language restriction: PubMed, Web of Science, Embase, Scopus, Cochrane Library and Google Scholar. Two independent reviewers performed study screening, data extraction, and quality assessment, with a third author resolving discrepancies. All original studies on the use of AI techniques in RE were identified and the effectiveness of these techniques was compared. A critical appraisal was conducted using the QUADAS-2 risk-of-bias tool. A meta-analysis was performed using R software (version 4.5.0). Of 6,288 records retrieved, 45 met eligibility for systematic review, with 19 included in meta-analysis. Among these 45 studies, 55.5% (25/45) applied deep learning (DL) approaches, while 44.4% (20/45) employed machine learning (ML) techniques. The pooled sensitivity, specificity, diagnostic odds ratio (DOR), and summary of receiver operating characteristic (SROC) for detection and/or diagnosis studies were 0.94 (95%CI, 0.90-0.97), 0.96 (95%CI, 0.92-0.98), 382.56 (95% CI 111.91 -1307.77) and 0.98 (95%CI, 0.91-0.97), respectively. For prediction of REs, the pooled sensitivity, specificity, DOR, and SROC were 0.87

**Data availability statement:** All relevant data and materials supporting the conclusion of this article is/are available within the manuscript and its supporting information files.

**Funding:** The author(s) received no specific funding for this work.

**Competing interests:** The authors have declared that no competing interests exist.

**Abbreviations:** AC: Alignment Curve; AL: Axial Length;AL/CR: Axial Length-To-Corneal Radius Ratio; AI: Artificial Intelligence; ANN, Artificial Neural Network; AUC, Area Under The Curve; CCT, Central Corneal Thickness; CNN, Convolutional Neural Network; CR, Corneal Radius; DNN, Deep Neural Network; DL, Deep Learning; DOR, Diagnostic Odds Ratio; ICL, Implantable Collamer Lens; K1, Keratometry 1; K2, Keratometry 2; LASIK, Laser-Assisted In Situ Keratomileusis; LASEK, Laser-Assisted Subepithelial Keratectomy; LD, Lens Diameter; LASSO, Least Absolute Shrinkage and Selection Operator; MAE, Mean Absolute Error; MSE, Mean Squared Error; ML, Machine Learning; OCT, Optical Coherence Tomography; Ortho-K, Orthokeratology; PRISMA, Preferred Reporting Items for Systematic Review and Meta-Analysis; R², Coefficient of Determination; SE, Spherical Equivalent; SER, Spherical Equivalent Refraction; SMILE, Small Incision Lenticule Extraction; SROC, Summary Receiver Operating Characteristic; SVM, Support Vector Machine; TP, Treatment Power; URE, Uncorrected Refractive Error; WTW, White-to-White Distance; XGBoost, Extreme Gradient Boosting. 2023.

(95%CI, 0.73-0.94), 0.96 (95%CI, 0.90-0.980), 159.94 (95% CI, 40.17-636.85) and 0.96 (95%CI, 0.85-0.95), respectively. Among studies focused on progression, performance metrics ranged from AUC = 0.845-0.99, R² = 0.613-0.964, and MAE = 0.119D-0.49D. In treatment studies, performance varied more widely, with AUC values between 0.60–0.94 and MAE from 0.17D-0.54D. Collectively, AI technologies, particularly DL and ML, achieved high diagnostic and predictive accuracy in RE management. Future research should focus on developing generalizable models trained on diverse datasets to ensure broad clinical relevance.

## Author summary

Refractive errors, such as nearsightedness, farsightedness, and astigmatism, are common vision problems that affect people of all ages. They can usually be corrected with glasses, contact lenses, or surgery, but early and accurate detection is key to effective treatment. In recent years, AI has become a valuable tool in improving how these conditions are diagnosed, monitored, and managed. We investigate how AI is being used to detect, predict, track progression, and help treat these conditions. We analyzed 45 studies, of which 19 were included in a detailed meta-analysis. Over half of the included studies used DL, a type of AI that mimics the human brain, while the others used ML, an AI approach where computers identify patterns from data. We found that AI tools were very accurate at diagnosing and predicting refractive errors, with high sensitivity (correctly identifying those with the condition) and specificity (correctly identifying those without it). Further, the AI was also useful in tracking changes in vision over time and in helping guide treatment choices, although performance varied more in those areas. Overall, AI shows strong potential to improve how we detect and manage refractive errors. Future work should focus on developing AI systems that work well across different populations and clinical settings.

## Background

Refractive errors (REs) are among the most common vision problems, affecting millions of people worldwide—from infancy to old age [1]. These REs, including myopia, hyperopia, and astigmatism, arise from structural abnormalities in the eye that prevent proper light focus on the retina, resulting in blurred vision [2,3]. According to the World Vision Report [4], over 2.2 billion people are visually impaired or blind, including 101.2 million with moderate to severe visual impairment and 6.8 million cases of blindness due to uncorrected refractive errors (UREs) [5]. If left untreated, UREs can significantly impact individuals and communities by reducing quality of life, limiting education and employment opportunities, and hindering economic productivity [6]. Visual impairments due to UREs are projected to rise [7], highlighting the urgent need for enhanced screening and management strategies for early detection [8].

Traditional methods for diagnosing and managing REs rely on several eye tests such as visual acuity testing [9], objective refraction [10] and subjective refraction [11]. Objective techniques such as autorefraction and retinoscopy offer quicker, more standardized alternatives. However, autorefraction, especially without cycloplegia, is prone to over-estimating myopia or underestimating hyperopia due to accommodation, while retinoscopy - though accurate, requires technically skilled personnel, which may not be feasible during high-volume screening. Subjective refraction requires patient input, which may be unreliable - particularly in pediatric populations [12,13], and individuals with intellectual disabilities [14]. Additionally, it can be time-consuming, especially in large-scale screenings [12,15]. Effective intervention requires an early detection, monitoring of REs to avert the associated pathological challenges [16]. However, the growing burden of UREs may overwhelm existing eye healthcare systems which calls for exploitation of optimally smart strategies [17].

Artificial intelligence (AI) enables computer systems to perform tasks requiring human intelligence, such as problem-solving, reasoning, learning, and decision-making [18,19]. Its primary goal is to enhance computational power and automate tasks with minimal human input, boosting productivity and efficiency [20]. AI has become a transformative technology with broad impacts across industries, including ophthalmology [21,22]. It has demonstrated strong effectiveness in diagnosing various eye conditions, such as age-related macular degeneration [23,24], cataract [25,26], diabetic retinopathy [27,28], and glaucoma [29].

Artificial intelligence encompasses technologies such as ML and DL [30]. Machine learning algorithms excel at analyzing complex, non-linear relationships between predictors and outcomes, improving predictive accuracy [31]. Similarly, DL algorithms are particularly effective at interpreting intricate ocular images, detecting subtle patterns that distinguish healthy from abnormal eyes [30]. In the context of REs, ML and DL have shown strong performance in detection [34], diagnosis [32,33], prediction [34], monitoring progression [35], and treatment [36] utilizing large datasets of images and clinical data [37,38]. These successes demonstrate AI's ability to extract critical features from complex ophthalmic data, enhancing accuracy, efficiency, and personalized vision care [39–41]. This systematic review and meta-analysis aims to provide comprehensive evidence from the current literature on AI applications in REs, with a focus on its roles in detection, diagnosis, prediction, monitoring, and treatment.

## Methods

This protocol was registered with PROSPERO (CRD42024512157). The systematic review and meta-analysis followed Preferred Reporting Items for Systematic Review and Meta-analysis (PRISMA) checklist for transparency of reporting (S1 PRISMA Checklist) [42] and adhered to methods outlined in the Cochrane Handbook for robustness and reproducibility [43]. The study selection process and database screening were systematically documented using the PRISMA flow diagram (**Fig 1**) [44].

### Eligibility criteria

We included original, full-text articles published in peer-reviewed English-language journals that applied AI techniques (e.g., DL and ML), to ophthalmic imaging and/or clinical data to address REs. Studies involving non-human subjects, case reports, case series, reviews, commentaries, editorials, and opinion pieces were excluded.

### Search strategy

A literature search was performed in PubMed, Web of Science, Embase, Scopus, Cochrane library and Google Scholar for peer-reviewed articles published up to January 2025. The search combined terms related to REs (e.g., myopia, hyperopia, astigmatism), AI (e.g., machine learning, deep learning), and management (e.g., diagnosis, prediction, treatment). Detailed search strings are provided in S2 Text.

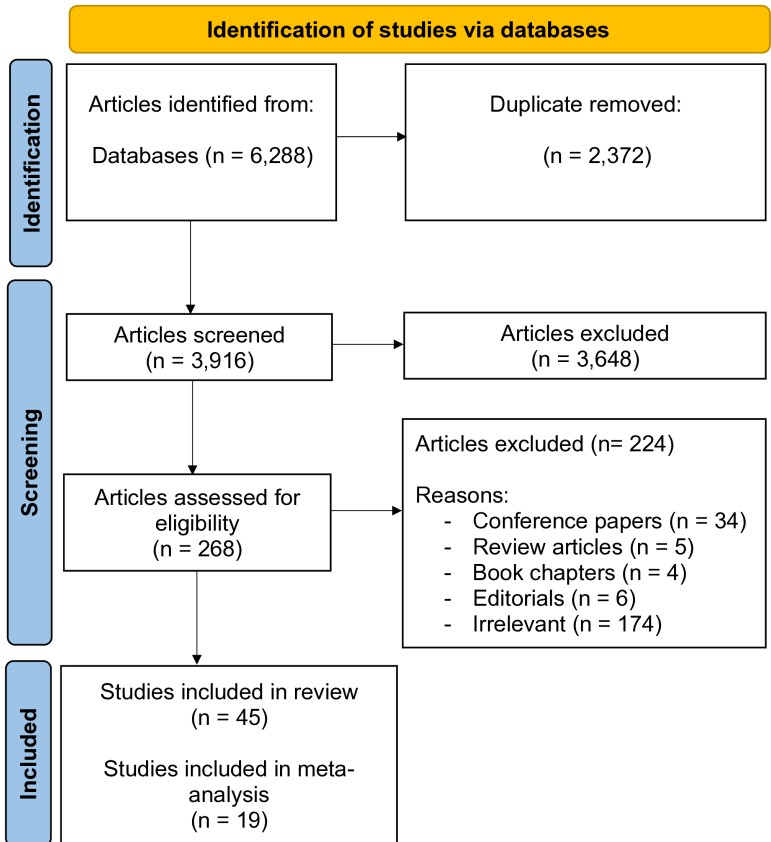

**Fig 1. PRISMA flow chart showing the study retrieval and study selection.**

## Study selection

All publications identified from the databases were imported into Covidence software [45] for processing, including duplicate removal, title and abstract screening, and full-text review. Two authors (JA and KAS) independently screened titles and abstracts based on predefined inclusion and exclusion criteria. Studies passing this stage underwent full-text review for final inclusion. Any disagreements were resolved through discussion between the reviewers. Additionally, reference lists of potentially relevant studies were examined to identify any missed articles.

## Data extraction

The primary outcome was to identify AI techniques used in RE diagnosis, detection, prediction, progression, and treatment, and to evaluate their effectiveness in RE management. Two authors (JA and EA) independently extracted data using a Covidence-designed extraction sheet. Extracted data included study characteristics (author, year, country, design, sample size), type of images or data, AI techniques and models used, performance metrics, and main outcomes. Discrepancies were resolved through discussion, with a third reviewer (SA) consulted for complex issues.

## Quality Assessment

Risk of bias was assessed using the QUADAS-2 tool, which evaluates four domains: patient selection, index test, reference standard, and flow and timing [46]. Each domain was rated as high, low, or unclear risk of bias. The first three

domains were also assessed for applicability concerns using the same rating scale. Signaling questions guided the evaluations and two authors (JA and SOP) independently performed the assessments, resolving disagreements through discussion and consensus.

### Data analysis

Quality assessment was performed using RevMan 5.3 (Cochrane Collaboration, Denmark), and meta-analysis was conducted in R (version 4.5.0). Studies were eligible for meta-analysis if they provided sufficient data to construct 2×2 diagnostic tables, or reported sensitivity and specificity values. For studies evaluating multiple algorithms, results from the highest-performing algorithm per dataset were selected. When 2×2 diagnostic tables were unavailable, calculations were derived from sensitivity, specificity, precision, or F1 scores along with case numbers. Meta-analysis grouped studies into detection and/or diagnosis and prediction categories, using random-effects models to synthesize data. Pooled analyses assessed diagnostic performance indicators such as sensitivity, specificity, DOR, and SROC. Study heterogeneity and threshold effects were also evaluated.

Heterogeneity among studies was assessed using the I² statistic, with values ≥50% indicating substantial heterogeneity [47]. Threshold effects were evaluated via Spearman's correlation between logit-transformed sensitivity and 1-specificity, with $p < 0.05$ deemed statistically significant. Subgroup analyses were planned based on input modalities and AI technique covariates. Publication bias was assessed using Egger's funnel plots and asymmetry tests. Progression and treatment studies were synthesized narratively.

### Dealing missing data

Only records with sufficient information for primary outcomes were included. Missing data were not imputed.

### Ethics and dissemination

As this study used secondary data, ethical approval was not required. Findings will be shared with stakeholders, presented at scientific conferences, published in a peer-reviewed journal, and disseminated via public social media platforms.

## Results

The literature selection process is summarized in a PRISMA flow diagram (**Fig 1**). The initial search identified 6,288 articles. After removing 2,372 duplicates, 3,916 titles and abstracts were screened, excluding 3,648 articles. This left 268 full-text articles for eligibility assessment, of which 224 were excluded (conference papers, reviews, book chapters, editorials, or irrelevant). Ultimately, 45 studies were included in the review, with 19 eligible for meta-analysis.

### Study characteristics

The characteristics of the 45 included studies are summarized in **Table 1**. These studies were published between 2013 and 2024. Most were conducted in Asia, with 26 (57.8%) in China and 8 (17.8%) in Korea/South Korea. The remainder originated from Singapore, India, New Zealand, the UK, and Croatia. Study designs were primarily retrospective (n = 22 studies), followed by cross-sectional (n = 3 studies), longitudinal (n = 4 studies), and one each of prospective [48], longitudinal cross-sectional [49], and retrospective clinical trial [50]. Thirteen studies did not specify their design, and regarding validation, 31 studies used internal validation only, while 14 applied both internal and external validation.

### AI applications

AI applications in RE management included detection and/or diagnosis (n = 14 studies), prediction (n = 15), progression monitoring (n = 5 studies), and treatment (n = 11 studies). Deep learning (DL) was the most used technique (n = 25 studies),

**Table 1. Characteristics of included studies.**

| Author, Year | Study Country | Study Design | Total sample size | | Type of Images/Data Used | AI Technique | Model used | Validation type | Model Aim |
|---|---|---|---|---|---|---|---|---|---|
| | | | Patients | Images/Data | | | | | |
| Li *et al*. 2023 [51] | China | Retrospective | 497 | N/A | Corneal topography maps | ML | Bagging Tree Gaussian Process SVM Decision Tree | Internal | Estimating the original corneal curvature after ortho-k |
| Varosanec *et al*. 2024 [52] | Croatia | Longitudinal | 895 | 10,170 | Clinical data | DL | RNN: Extended gate time-aware long short-term memory | Internal | Predicting future spherical equivalent |
| Li *et al*. 2024a [53] | China | Retrospective | 227,543 | 612,530 | Clinical data | ML | Multivariate linear regression Logistic regression | Internal External | Predicting the progression of myopia and progression to high myopia |
| Lu *et al*. 2021a [54] | China | Retrospective | 28,913 | 32,010 | Fundus images | DL | CNN: ResNet-18 Faster R-CNN | Internal External | Identifying non-pathologic myopia and pathologic myopia |
| Jiang *et al*. 2023 [55] | China | Retrospective | 1678 | 2767 | Clinical data | ML | Support vector regression Random Forest XGBoost LASSO | Internal | Predicting postoperative refraction errors |
| Pathan *et al*. 2020 [56] | N/A | N/A | N/A | 400 | Fundus images | ML | Multilayer perceptron AdaBoost | Internal External | Detecting pathological myopia and non-pathological myopia |
| Rauf *et al*. 2021 [57] | N/A | N/A | N/A | 400 | Fundus images | DL | CNN | Internal External | Detecting pathological myopia |
| Zhang *et al*. 2013 [58] | China | Cross-sectional | 2,258 | N/A | Fundus images Clinical data Genotyping data | ML | SVM | Internal | Detecting pathological myopia |
| Peng *et al*. 2024 [59] | China | N/A | 2492 | N/A | Fundus images | DL | Attention-based Patch Residual Shrinkage network | Internal | Diagnosing paediatric high myopia |
| Yang *et al*. 2019 [60] | China | Retrospective | N/A | 2350 | Ocular appearance images | DL | DCNN: VGG-Face | Internal | Identifying the presence of myopia |
| Linde *et al*. 2023 [61] | New Zealand India | Retrospective | 512 | N/A | Pupillary red reflex images | DL | CNN: Inception-V3 EfficientNet | Internal | Estimating refractive error |
| Xu *et al*. 2022 [62] | China | N/A | N/A | 3103 | Photorefraction images | DL | CNN+RNN | Internal | Predicting refractive error |
| Varadarajan *et al*. 2018 [63] | United Kingdom | N/A | 64755 | N/A | Fundus images | DL | ResNet | Internal | Predicting refractive error |
| Yoo *et al*. 2021 [64] | South Korea | Retrospective | 468 | 936 | OCT images | DL | ResNet50 InceptionV3 VGG16 | Internal External | Estimating uncorrected refractive error |
| Park *et al*. 2022 [65] | Korea | Retrospective | 367 | 36700 | OCT images | DL | CNN: ResNet18 ResNext50 EfficientNetB0 EfficientNetB4 | Internal | Distinguishing between pathological myopia group and normal group |
| Choi *et al*. 2021 [66] | Korea | Retrospective | 436 | 1,200 | OCT images | DL | CNN: VGG 16 ResNet 50 Inception V3 | Internal | Distinguishing high myopia from normal and other retinal diseases |

*(Continued)*

**Table 1.** (Continued)

| Author, Year | Study Country | Study Design | Total sample size | | Type of Images/Data Used | AI Technique | Model used | Validation type | Model Aim |
|---|---|---|---|---|---|---|---|---|---|
| | | | Patients | Images/ Data | | | | | |
| Foo et al. 2023 [67] | Singapore | Retrospective | 965 | 7456 | Fundus images Clinical data | DL | DNN: DenseNet-121 Random Forest | Internal External | Predicting the development of high myopia |
| Chun et al. 2020 [68] | Korea | N/A | 164 | 305 | Photorefraction images | DL | CNN: ResNet-18 | Internal | Predicting the range of refractive error |
| Zhang et al. 2024 [69] | China | Retrospective | 369 | 1346 | Corneal topography maps | DL | DNN: Segformer Architecture Network | Internal | Determining the Treatment Zone and Peripheral Steepened Zone following ortho-K |
| Jain et al. 2024 [70] | South Korea India | Cross-sectional | 1331 | 2662 | OCT images | DL | CNN: ResNet50 | Internal | Predicting uncorrected refractive error |
| Yang et al. 2024 [71] | China | Retrospective | 266 | 449 | Clinical data | DL | DNN: Dense | Internal | Predicting lens prescription parameters in ortho-K |
| Koo et al. 2024 [72] | Korea | Retrospective | 297 | 547 | Clinical data | ML | Decision tree CatBoost Extra Trees XGBoost Random Forest Least Angle regression Ridge LASSO | Internal | Selecting ortho-K lens parameters |
| Lu et al. 2021b [73] | China | Retrospective | 13869 | 17330 | Fundus images | DL | CNN: DenseNet201 ResNet50 VGG16 Xception | Internal External | Detecting pathological myopia |
| Peng et al. 2023 [74] | China | Prospective | 2538 | N/A | Clinical data | ML | Random Forest SVM Gradient Boosting Decision Tree CatBoost | Internal | Predicting the onset of myopia |
| Ren et al. 2023 [75] | China | Retrospective | N/A | 1156 | Fundus images | DL | ResNet18 Faster R-CNN | Internal | Diagnosing pathological myopia |
| Yoo et al. 2020 [76] | Korea | Retrospective | 18,480 | N/A | Clinical data | ML | XGBoost SVM Random forest ANN | Internal External | Selecting best laser refractive surgery option |
| Fan et al. 2022 [77] | China | Retrospective | 1,271 | N/A | Clinical data | ML | Linear regression SVM Bagging decision trees Gaussian processes | Internal | Estimating alignment curve curvature in ortho-K lens fitting |
| Fang et al. 2023 [50] | China | Retrospective clinical trial | 91 | N/A | Clinical data | ML | LASSO regression | Internal | Predicting the treatment effect of ortho-k |
| Xu et al. 2023 [78] | China | Retrospective | 1,302 | N/A | Clinical data | ML | SVM Gaussian process regulator Decision tree Random Forest | Internal | Predicting ortho-K lens parameters and axial length progression |

*(Continued)*

| Author, Year | Study Country | Study Design | Total sample size | | Type of Images/Data Used | AI Technique | Model used | Validation type | Model Aim |
|---|---|---|---|---|---|---|---|---|---|
| | | | Patients | Images/Data | | | | | |
| Ying *et al.* 2024 [31] | China | Cross-sectional | 3,414 | 6,827 | Clinical data | ML | SVM<br>Random Forest<br>XGBoost<br>MLP-NN<br>Linear regression<br>LASSO regression | Internal External | Predicting cycloplegic SER and myopia status |
| Kim *et al.* 2021 [79] | South Korea | Retrospective | 1839 | N/A | Fundus images Clinical data | ML | SVM<br>Decision Tree<br>Random Forest<br>K-nearest neighbors<br>Naïve Bayes classifiers | Internal | Predicting pathological myopia |
| Huang *et al.* 2023 [80] | China | Retrospective | 37586 | 75172 | Clinical data | DL | RNN: T-LSTM<br>Standard LSTM<br>Random Forest<br>Linear Regression | Internal | Predicting myopia |
| Zhao *et al.* 2024a [81] | China | N/A | N/A | 7,114 | Fundus images | DL | CNN: ResNet-101 | Internal | Classifying pathological myopia |
| Hemelings *et al.* 2021 [82] | N/A | N/A | N/A | 1200 | Fundus images | DL | ResNet-18 U-Net++ | Internal External | Detecting and classifying pathological myopia |
| Yang *et al.* 2020 [83] | China | N/A | 3112 | N/A | Clinical data | ML | SVM | Internal | Studying influence of related factors to predict myopia |
| Barraza-Bernal *et al.* 2023 [84] | China | Cross-sectional | 12780 | N/A | Clinical data | ML | Gaussian process regression<br>Support vector regression<br>Support vector machine (SVM) | Internal | Predicting refractive error its development over time |
| Yang *et al.* 2022 [85] | China | N/A | 987 | 987 | Fundus images | DL | ResNet-50<br>Inception-v3<br>Inception-ResNet-v2 | Internal External | Predicting refractive error in myopic patients |
| Yuan *et al.* 2023 [86] | China | N/A | N/A | 14,028 | Clinical data | ML | BP neural network | Internal | Predicting cutting for-mula of SMILE |
| Li *et al.* 2024b [87] | China | Longitudinal | 12,766 | 12,766 | Clinical data | ML | XGBoost<br>K Neighbors<br>Decision tree<br>Logistic regression<br>Gaussian NB | Internal | Predict myopia progression and the risk of developing high myopia |
| Zhu *et al.* 2023 [88] | China | Retrospective | N/A | 179 | Clinical data | ML | Orthogonal matching pursuit<br>Random Forest<br>Kernel ridge regression<br>K-nearest neighbor regression<br>Extra tree regression<br>Multilayer perceptron | Internal | Predict the changes in SER and AL |
| Tan *et al.* 2021 [89] | Singapore | Retrospective | 112,110 | 225,671 | Fundus images | DL | CNN: ResNet-101 | Internal External | Detecting high myopia |
| Ali and Raut. 2024 [90] | N/A | N/A | N/A | 400 | Fundus images | DL | CNN: Spatial attention network<br>Squeeze- excitation network | Internal External | Detecting pathological myopia |

*(Continued)*

**Table 1.** (Continued)

| Author, Year | Study Country | Study Design | Total sample size Patients | Total sample size Images/Data | Type of Images/Data Used | AI Technique | Model used | Validation type | Model Aim |
|---|---|---|---|---|---|---|---|---|---|
| Zhao *et al.* 2024 b [91] | China | Longitudinal | 88,250 | 408,255 | Clinical data | ML | Random forest XGBoost | Internal | Predict SE and development of myopia and high myopia |
| Lin *et al.* 2018 [92] | China | Longitudinal | 129242 | 687063 | Clinical data | ML | Random forest | Internal External | Predict the onset of high myopia, at specific future time points |
| Tang *et al.* 2021 [93] | China | N/A | 2044 | 6328 | Corneal topography maps | DL | DNN+CNN | Internal | Identify treatment zone boundary and treatment zone center |

DL Deep learning; ML Machine learning; CNN Convolutional neural network; RNN Residual neural network; DNN Deep neural network; DCNN Deep convolutional neural network; SMILE Small incision lenticule extraction; SVM Support vector machine; Ortho-K Orthokeratology

followed by machine learning (ML) in 20 studies. Common DL architectures included ResNet (n = 13 studies), Inception V3 (n = 4 studies), and VGG Face (n = 4 studies). Among ML models, support vector machine (SVM) (n = 9 studies), random forest (n = 10 studies), and XGBoost (n = 6 studies) were prevalent. AI methods analyzed retinal fundus images (n = 15 studies), OCT images (n = 4 studies), clinical data (n = 22 studies), eccentric photorefraction images (n = 4 studies), corneal topography maps (n = 2 studies), ocular appearance images (n = 1 study), genotyping data (n = 1 study), or combinations thereof (See **Table 1** and **Fig 2**).

## Risk of bias of studies

The included studies were of moderate to high quality. Forty-one studies (91.1%) were rated as low risk of bias across all four QUADAS-2 domains. In patient selection, 4 studies (8.8%) had unclear risk and applicability concerns due to insufficient dataset descriptions. Most studies (91.1%) showed low risk and concerns in the index test domain, with 4 studies (8.8%) marked unclear due to data overlap. All studies had low risk in the reference standard domain and for flow and timing, 3 studies (6.7%) were rated unclear because of poor documentation on dataset assembly (see S3 Table).

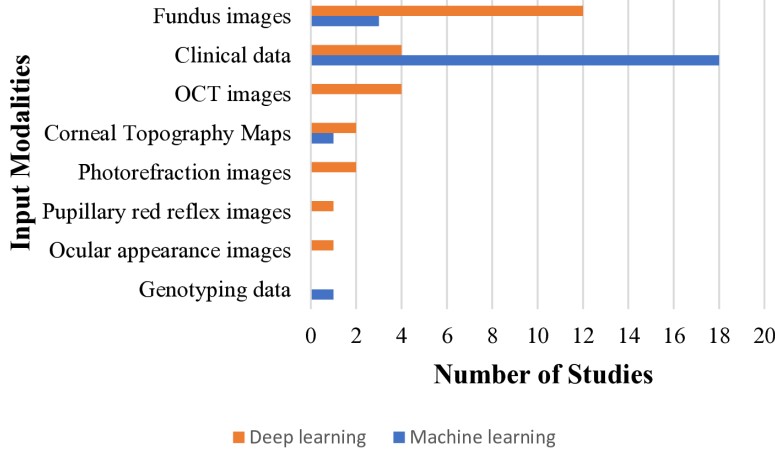

**Fig 2. Types of AI techniques and input modalities used in the included studies.**

## AI in refractive error detection and diagnosis

Out of 14 studies on detection and/or diagnosis of refractive errors, 11 were included in the meta-analysis. The pooled sensitivity was 0.94 (95% CI: 0.90–0.97) and specificity 0.96 (95% CI: 0.92–0.98) (see **Fig 3**). The DOR was 382.56 (95% CI: 111.91–1307.77) and the SROC was 0.98 (95% CI: 0.91–0.97) as shown in Figs 4 and 5.

## Threshold analysis for detection and diagnosis studies

Threshold analysis showed no significant correlation (Spearman r = -0.536, p = 0.094), indicating that variability among studies was unlikely due to differing diagnostic thresholds.

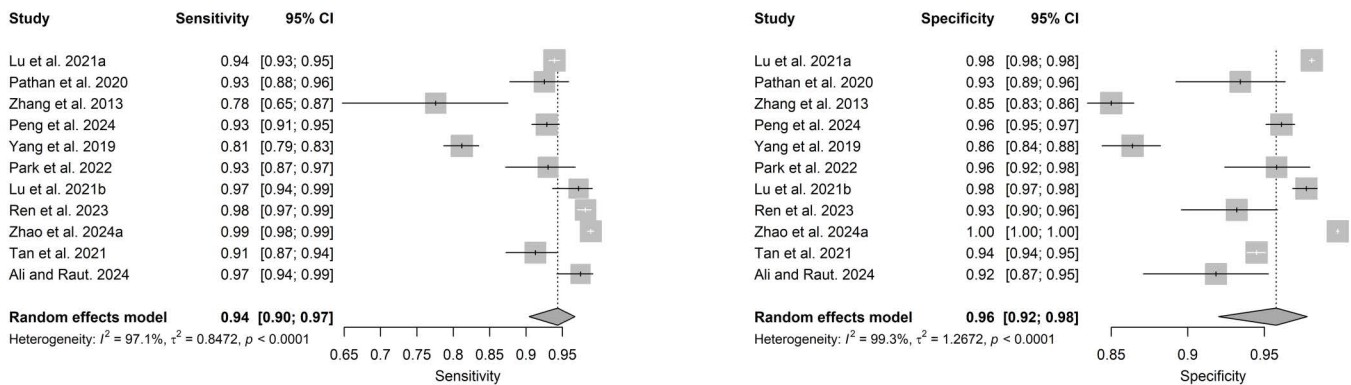

**Fig 3. Forest plots for sensitivity (3A) and specificity (3B) for AI in refractive error detection or diagnosis.**

| Study | DOR | 95% CI | Odds Ratio |
|---|---|---|---|
| Lu et al. 2021a | 793.94 | [ 679.40; 927.78] | |
| Pathan et al. 2020 | 175.65 | [ 81.47; 378.72] | |
| Zhang et al. 2013 | 19.62 | [ 10.47; 36.76] | |
| Peng et al. 2024 | 320.61 | [ 222.06; 462.89] | |
| Yang et al. 2019 | 27.36 | [ 21.93; 34.15] | |
| Park et al. 2022 | 304.00 | [ 120.27; 768.39] | |
| Lu et al. 2021b | 1515.61 | [ 584.04; 3933.02] | |
| Ren et al. 2023 | 736.38 | [ 373.31; 1452.58] | |
| Zhao et al. 2024a | 42944.40 | [24623.90; 74895.60] | |
| Tan et al. 2021 | 179.74 | [ 115.38; 280.00] | |
| Ali and Raut. 2024 | 434.25 | [ 155.88; 1209.71] | |
| **Random effects model** | **382.56** | **[ 111.91; 1307.77]** | |

Heterogeneity: $I^2 = 99.0\%$, $\tau^2 = 4.2105$, $p < 0.0001$

Diagnostic Odds Ratio: 0.001  0.1  1  10  1000

**Fig 4. Forest plot for DOR for detection or diagnosis studies.**

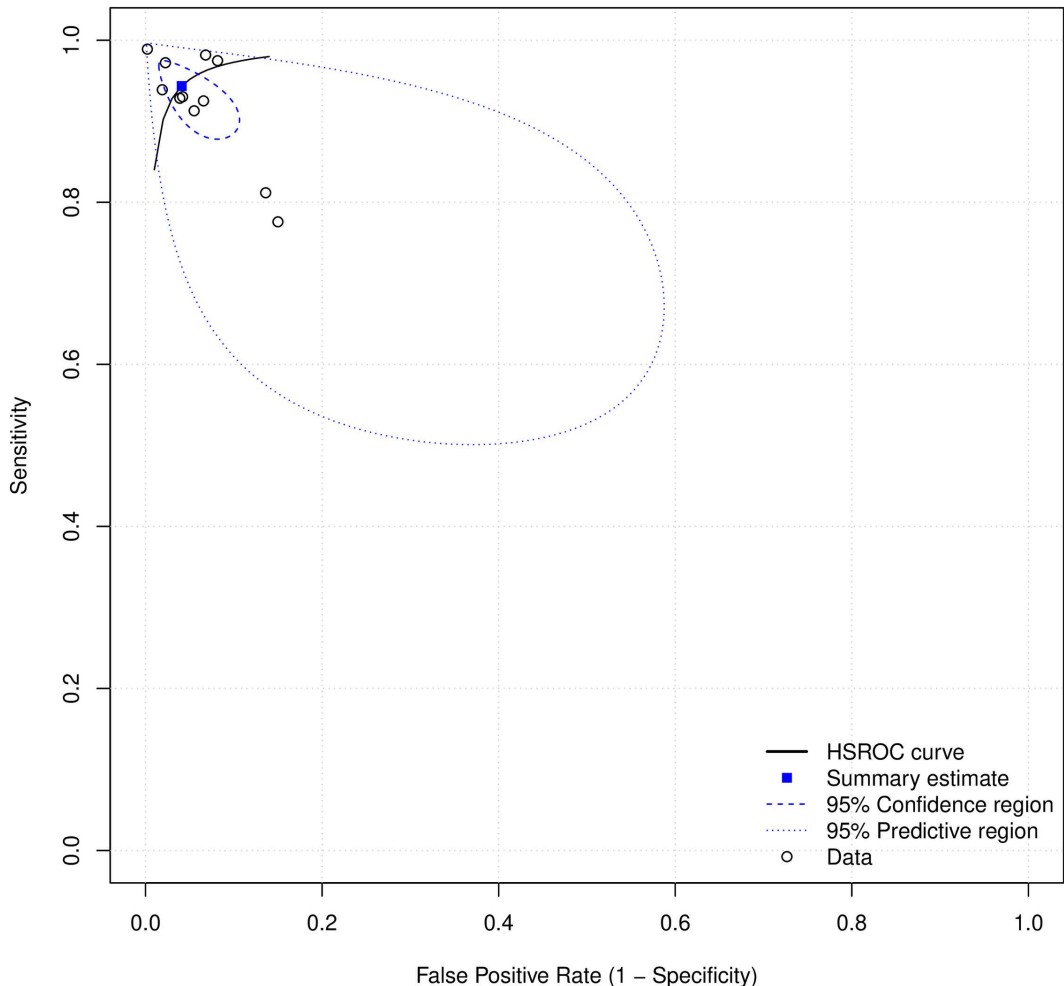

**Fig 5. SROC curve for detection or diagnosis studies.**

## Publication bias for detection and diagnosis studies

Egger's funnel plot asymmetry test showed no publication bias (P = 0.902), as illustrated in S4 Fig.

## Subgroup analysis by AI technique for detection and diagnosis studies

Nine studies used DL as the AI model backbone, with pooled sensitivity of 0.95 (95% CI: 0.92–0.97), specificity 0.97 (95% CI: 0.93–0.98), and DOR 580.30 (95% CI: 147.92–2276.57). The remaining two studies used ML, showing pooled sensitivity of 0.87 (95% CI: 0.65–0.96), specificity 0.90 (95% CI: 0.78–0.95), and DOR 58.02 (95% CI: 6.77–497.17). A significant difference was found in pooled specificity (p = 0.04) between DL and ML, but no significant differences were observed for sensitivity (p = 0.11) or DOR (p = 0.08) –see S5 Fig.

## Subgroup analysis by input modalities for detection and diagnosis studies

Eight studies used fundus images, while one study each used optical coherence tomography (OCT) images, ocular appearance images, and a combination of fundus images, clinical data, and genotyping data. For fundus images, pooled

sensitivity was 0.96 (95% CI: 0.94–0.98), specificity 0.97 (95% CI: 0.95–0.98), and DOR 801.47 (95% CI: 286.63–2241.08). The ocular appearance image study showed sensitivity of 0.81 (95% CI: 0.79–0.83), specificity of 0.86 (95% CI: 0.84–0.88), and DOR of 27.36 (95% CI: 21.93–34.15). The OCT image study reported sensitivity of 0.93 (95% CI: 0.87–0.97), specificity of 0.96 (95% CI: 0.92–0.98), and DOR of 304.00 (95% CI: 120.27–768.39). The combined modality study yielded sensitivity of 70.8 (95% CI: 0.65–0.87), specificity of 0.85 (95% CI: 0.83–0.86), and DOR of 19.62 (95% CI: 10.47–36.76). Significant differences in pooled sensitivity, specificity, and DOR were observed between input modalities (p < 0.0001)- see S6 Fig.

## AI in refractive error prediction

Eight of 14 studies on RE prediction were included in the meta-analysis. The pooled sensitivity was 0.87 (95% CI: 0.73–0.94), specificity 0.96 (95% CI: 0.90–0.98), DOR 159.94 (95% CI: 40.17–636.85), and SROC 0.96 (95% CI: 0.85–0.95) as shown in .

## Threshold analysis for prediction studies

Threshold analysis showed no significant correlation (Spearman r = -0.119, p = 0.793), indicating heterogeneity among studies was unlikely due to differences in diagnostic thresholds.

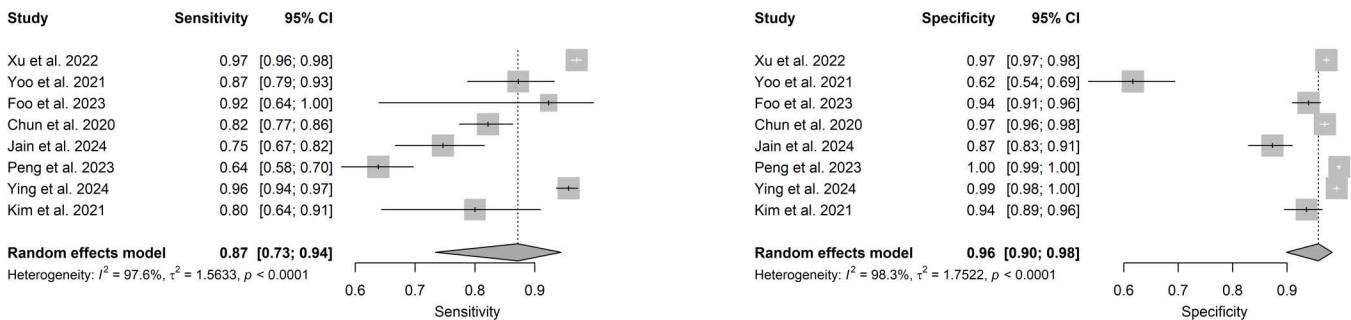

**Fig 6. Forest plots for sensitivity (6A) and specificity (6B) for AI in refractive error prediction.**

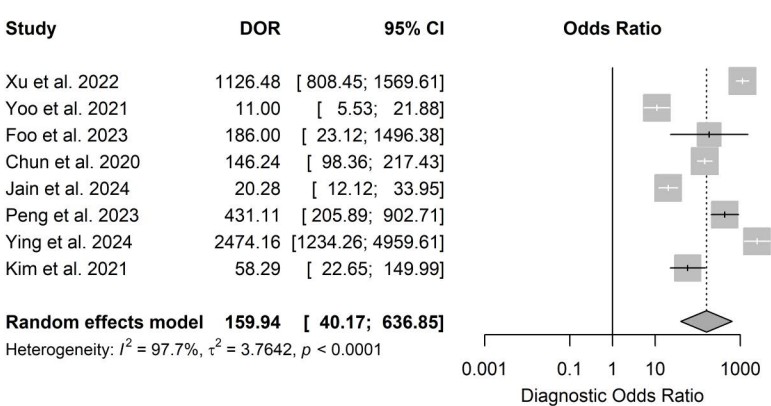

**Fig 7. Forest plot for DOR for prediction studies.**

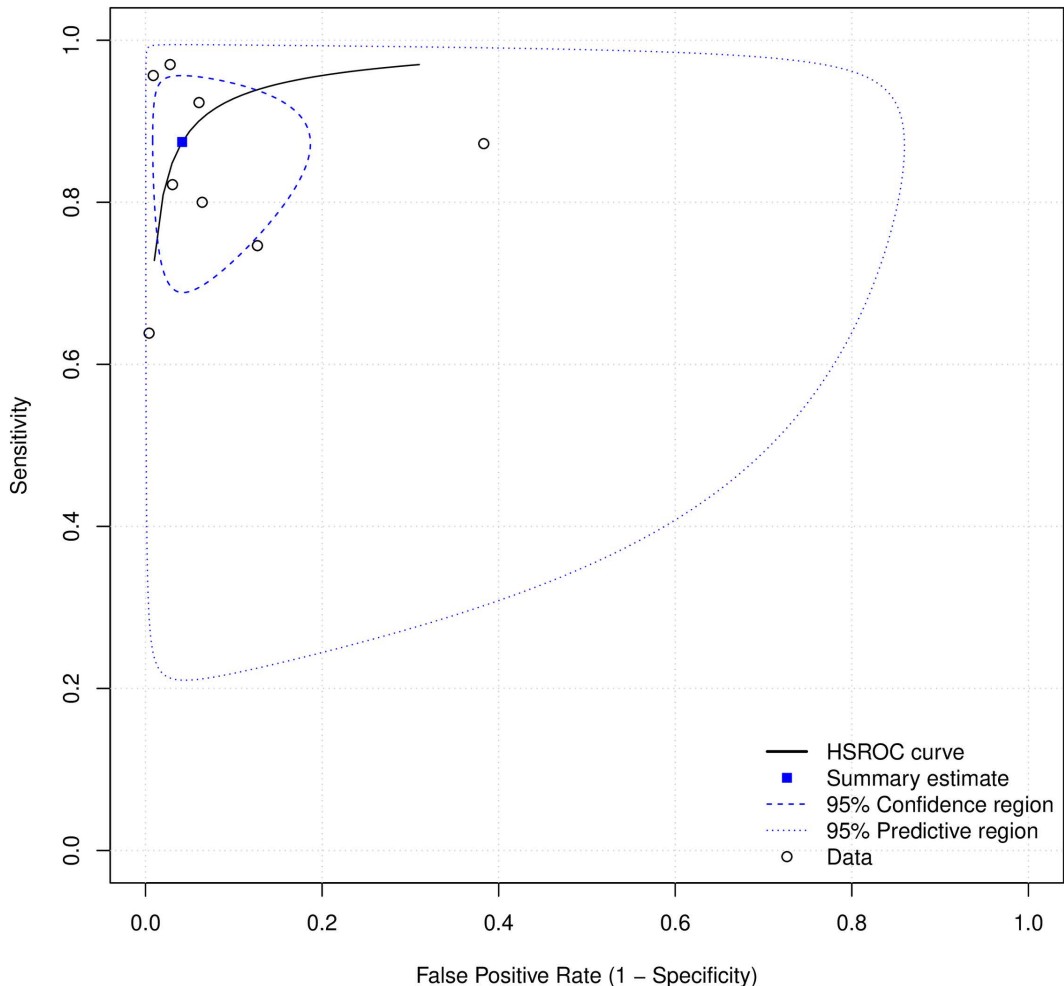

**Fig 8. SROC curve for prediction studies.**

### Publication bias for prediction studies

Egger's funnel plot asymmetry test showed no publication bias (p = 0.495), as illustrated in S7 Fig).

### Subgroup analysis by AI technique for prediction studies

Five studies used DL as the backbone, with a pooled sensitivity of 0.89 (95% CI, 0.72–0.96), slightly outperforming the three studies using ML, which had a sensitivity of 0.84 (95% CI, 0.47–0.97); however, the difference was not statistically significant (p = 0.72). ML studies demonstrated higher pooled specificity (0.99, 95% CI, 0.93–1.00) and DOR (404.52, 95% CI, 53.21–3075.36) compared to DL studies (specificity 0.92, 95% CI, 0.79–0.97; DOR 91.07, 95% CI, 14.1–588.19), though these differences were also not statistically significant (p = 0.07 and p = 0.29, respectively) -see S8 Fig.

### Subgroup analysis by input modalities for prediction studies

Three studies used OCT images, two used eccentric photorefraction images, two used clinical data, and one study utilized a combination of fundus images and clinical data as input modalities. Eccentric photorefraction images demonstrated the

highest pooled sensitivity at 0.92 (95% CI, 0.64–0.99), followed by the combination of fundus and clinical data at 0.92 (95% CI, 0.64–1.00), clinical data at 0.86 (95% CI, 0.34–0.99), and OCT images at 0.81 (95% CI, 0.71–0.88). However, studies using clinical data alone had the highest pooled specificity (0.99, 95% CI, 0.99–1.00) and DOR (1037.61, 95% CI, 187.24–5750.0). These values were superior to those observed in studies using eccentric photorefraction images (specificity: 0.97, DOR: 407.1), OCT images (specificity: 0.84, DOR: 22.08), and the combined modality (specificity: 0.94, DOR: 186.00). The differences in specificity and DOR among input modalities were statistically significant ($p < 0.0001$) -see S9 Fig.

## Review of AI in refractive error treatment

A total of 11 studies investigated the application of AI in the treatment of REs. Due to the limited number of eligible studies and the heterogeneity in reporting styles, a meta-analysis was not feasible. The studies primarily addressed areas such as orthokeratology (Ortho-K), laser refractive surgeries (including LASEK: Laser-Assisted Subepithelial Keratectomy, LASIK: Laser-Assisted In Situ Keratomileusis, and small lenticule incision extraction (SMILE)), myopia control outcomes, and implantable collamer lens (ICL) implantation. All studies utilized either clinical data (n = 8) or corneal topography maps (n = 3) as input modalities.

Tree-based ML algorithms, such as decision trees, random forest, XGBoost, CatBoost, and bagging trees, were the most applied modeling techniques. Other ML models included support vector machines (SVM), Gaussian process regression, Least Absolute Shrinkage and Selection Operator (LASSO) regression, and artificial neural networks (ANN). Among DL techniques, full convolutional networks, and innovative architectures like Segformer were used for image-based analysis.

Input features across studies frequently included demographic and biometric parameters such as age, axial length (AL), keratometry values (K1, K2), spherical equivalent refraction (SER), and white-to-white (WTW) distance. Target variables varied and encompassed treatment zone characteristics (e.g., alignment, curvature, diameter), axial length changes, effectiveness of myopia control, and lens design parameters (e.g., base curve, lens sag, optical zone diameter).

Several studies demonstrated strong predictive performance using AI models in the treatment of REs. For example, Li *et al*. employed bagging tree algorithms to predict corneal curvature parameters (K1 and K2) following ortho-k, achieving R values ranging from 0.812 to 0.837 and mean absolute errors (MAEs) between 0.669 and 0.701 [51]. Yang *et al*. utilized a deep neural network (DNN) to predict lens design parameters alignment curve (AC), treatment power (TP), and lens diameter (LD)—with excellent results ($R^2$ up to 0.97; mean squared error (MSE) between 0.01 and 0.08) [71]. Koo *et al*. [72] implemented multiple ML models, including CatBoost and Extra Trees, and achieved high classification accuracy for toric lens recommendations (accuracy = 0.927; F1-score = 0.929). They also reported strong regression performance for lens base curve prediction ($R^2 = 0.948$).In another study, Fang *et al*. reached a classification accuracy of 92.86% for predicting myopia control outcomes, with a sensitivity of 86.67% and specificity of 100% [50].

In the context of refractive surgery planning, several AI-based approaches have shown promising results. Yoo *et al*. applied the XGBoost algorithm to recommend appropriate surgical procedures, achieving an accuracy of 82.1% [76] while Yuan *et al.* utilized a backpropagation (BP) neural network to estimate lenticular thickness for SMILE surgery, reporting a MSE of 0.248 [86]. Advanced image-based segmentation techniques were also explored. Zhang *et al.* [69] and Tang *et al*. [93] implemented DL architectures to delineate treatment zones on corneal topography maps with high precision—achieving a mean Intersection over Union (mIoU) of 97.19%, Dice Similarity Coefficient (DSC) of 0.94, and Intersection over Union (IoU) of 0.90 ± 0.06. For predicting postoperative refractive outcomes in ICL implantation, Jiang *et al.* [55] employed various ML models—including support vector regression, Random Forest, XGBoost, and LASSO regression. Using a comprehensive set of clinical parameters, they predicted postoperative SE and sphere values with strong accuracy, reporting MAEs of 0.339 diopters (D) for non-toric ICL (NT-ICL) and 0.325 D for toric ICL (TICL) lenses (see **Table 2**).

**Table 2. Characteristics of studies for refractive error treatment.**

| Author, Year | Type of Images/ Data Used | Input features/ (No.) | Treat-ment approach | AI Technique | Best model | Target variable | Performance metrics |
|---|---|---|---|---|---|---|---|
| Li *et al.* 2023 [51] | Corneal topog-raphy maps | Total number = 6 MTDP, K1after, K2after, K1af-ter_axis, K2after_axis, SER | Ortho-K | ML | Bagging Tree | K1 and K2 | K1 Model 1 R = 0.812, RMSE = 0.855, MAE = 0.671 K1 model 2 R = 0.812, RMSE = 0.858, MAE = 0.669 K2 Model 1 R = 0.831, RMSE = 0.898, MAE = 0.701 K2 Model 2 R = 0.837, RMSE = 0.888, MAE = 0.683 |
| Jiang *et al.* 2023 [55] | Clinical data | Total number = 20 Age, sphere, cylinder, and cylinder axis, IOP, mesopic PD, scotopic PD, AL, K1, K2, K1 axis, K2 axis, ACA, ACD, CT, WTW, NT-ICL lens sphere, TICL lens sphere, cylinder and lens axis | ICL implanta-tion | ML | Random forest | Postop- SE and sphere prediction of NT-ICL | NT-ICL-SE prediction MAE = 0.339D, SD = 0.445D, MedAE = 0.268D, Interquartile AE = 0.372D NT-ICL-sphere prediction MAE = 0.386D, SD = 0.489D, MedAE = 0.336D, Interquartile of AE = 0.312D |
| | | | | | XGBoost | Postop- SE and sphere prediction of TICL | TICL-SE prediction MAE = 0.325D, SD = 0.452D, MedAE = 0.257D, Interquartile of AE = 0.316D TICL-sphere prediction MAE = 0.308D, SD = 0.443D, MedAE = 0.241D, Interquartile of AE = 0.344D |
| Zhang *et al.* 2024 [69] | Corneal topog-raphy maps | N/A | Ortho-K | DL | Seg-former Archi-tecture Network | Treatment Zone (TZ) Peripheral Steepened Zone (PSZ) | ACC = 99.03%, mIoU = 97.19%, mPA = 98.98% |
| Yang *et al.* 2024 [71] | Clinical data | Total number = 9 Flat K, Steep K, Corneal astig-matism, Flat, Steep e, E mean, BFS, Sagittal differential at 8 mm corneal zone, HVID | Ortho-K | DL | DNN | Alignment Curvature (AC) Target Power (TP) Lens Diame-ter (LD) | For AC, TP and LD MSE = 0.08 D, 0.07 D and 0.01 mm R² = 0.97 D, 0.95 D and 0.91 mm |
| Koo *et al.* 2024 [72] | Clinical data | Total number = 16 Age, AR Sph and AR Cyl, MR Sph and MR Cyl, flattest and steepest keratometry, mean keratometry, keratometry astigmatism, WTW, e value, flat e value, steep e value, axial length, CCT, ACD | Ortho-K | ML | CatBoost | Toric option RZD2 LZA | 1. Toric lens: ACC = 0.927, Precision = 0.931, Recall = 0.927, F1 = 0.929 2. RZD2: MAE = 3.775 μm, RMSE = 11.703 μm, R² = 0.791 6. LZA: MAE = 0.121°, RMSE = 0.346°, R² = 0.798 |
| | | | | | Extra Trees | OAD LensSag | 1. OAD: ACC = 0.864, Precision = 0.868, Recall = 0.864, F1 = 0.865 2. LensSag:MAE = 4.372 μm, RMSE = 6.008 μm, R² = 0.921 |
| | | | | | Least-angle regres-sion | BC RZD1 | 1. BC: MAE = 0.054 μm, RMSE = 0.083μm, R² = 0.948 2. RZD1: MAE = 3.031 μm, RMSE = 8.574 μm, R² = 0.708 |
| Yoo *et al.* 2020 [76] | Clinical data | Total number = 9 Corrected distance VA, Mani-fest refraction, Slit-lamp exam-ination, and Dilated fundus examination, Corneal topog-raphy map, Central corneal thickness (CCT), Pupil size, NIBUT, Questionnaire survey | Refractive surgery | ML | Multi-class XGBoost | LASEK LASIK SMILE Contraindica-tion | ACC = 82.1% |

*(Continued)*

**Table 2.** (Continued)

| Author, Year | Type of Images/ Data Used | Input features/ (No.) | Treatment approach | AI Technique | Best model | Target variable | Performance metrics |
|---|---|---|---|---|---|---|---|
| Fan *et al.* 2022 [77] | Clinical data | Total number = 8 Gender, Age, HVID, SER, e value, flat K (K1), steep K (K2), ACD, AL | Ortho-K | ML | Linear SVM | Steep K reading of AC1 | AC1K1 R²= 0.91, MAE = 0.263, RMSE = 0.373, MSE = 0.139 |
| | | | | | Gaussian process regression | Flat K reading of AC1 Flat K reading of AC2 | AC1K2 R²= 0.84, MAE = 0.396, RMSE = 0.532, MSE = 0.283 AC2K1 R²= 0.73, MAE = 0.507, RMSE = 0.680, MSE = 0.462 |
| Fang *et al.* 2023 [50] | Clinical data | Total number = 9 Age, Baseline AL, Pupil diameter, Lens wearing time, Time spent outdoors, Time spent on near work, WTW, Anterior corneal flat K, Posterior corneal astigmatism | Ortho-K | ML | LASSO regression | Myopia control effect of ortho-k | ACC = 92.86% SEN = 86.67% SPE = 100% |
| Xu *et al.* 2023 [78] | Clinical data | Total number = 9 Age of myopia onset, Number of myopic parents, Spherical power, cylindrical power, Flat K, Steep K, Corneal diameter, Eccentricity value, Baseline Axial length | Ortho-K | ML | Random forest | AC and TP Axial length after 1 year | R = 0.97 R²= 0.93 MAE = 0.185 MSE = 0.093 |
| Yuan *et al.* 2023 [86] | Clinical data | Total number = 4 SPH, CYL, Km and lenticule diameter | SMILE surgery | ML | BP neural network | Lenticular thickness | MSE = 0.248 Gradient = 4.23 |
| Tang *et al.* 2021 [93] | Corneal topography maps | Total number = 2 Axial subtractive maps, Tangential subtractive maps | Ortho-K | DL | Full convolutional network | Treatment zone boundary Treatment zone center | Treatment zone boundary IoU = 0.90 ± 0.06 DSC of 0.94 ± 0.04 Treat zone center Deviation = 6.32 ± 6.23 pixels |

ML machine learning; DL Deep learning; DNN Deep neural network; SVM Support vector machine; Ortho-K Orthokeratology; K Keratometry; K1 Original flat K; K2 Original steep K; SER Spherical equivalent refraction; SE Spherical equivalent; AL Axial length; AR Autorefraction; MR Manifest refraction; ICL Implantable collamer lens; PD Pupillary distance; IOP Intraocular pressure; MTDP maximal tangential difference power; ACD Anterior chamber depth; ACA Anterior chamber depth; CT Corneal thickness; CCT Central corneal thickness; WTW white-to-white; NT-ICL non-toric ICL; TICL Toric-ICL; TZ Treatment Zone; PSZ Peripheral Steepened Zone; AC Alignment Curvature; TP Target Power; LD +Lens Diameter

## Review of progression of refractive error over a number of years

Among the included studies, six focused on predicting the progression of REs. Due to substantial heterogeneity in reporting styles and outcome measures, a meta-analysis was not feasible. These studies primarily investigated clinical outcomes related to RE progression trajectories, particularly the onset of myopia and high myopia. All six studies employed ML approaches, with commonly used algorithms including support vector machines (SVM), extreme gradient boosting (XGBoost), random forests, multivariate linear regression, and logistic regression. These models were trained on clinical datasets comprising input features such as age, SE, annual myopia progression rate, axial length-to-corneal radius (AL/ CR) ratio, time intervals between baseline and follow-up visits, and SE at subsequent follow-ups [53,91,92]. XGBoost demonstrated superior performance in two studies [87,91]. For example, Li *et al.* reported an AUC of 0.96 for predicting annual myopia progression [87] while Zhao *et al.* achieved high predictive accuracy across multiple outcomes (R²= 0.613– 0.992 for SE trajectory prediction and AUC = 0.845–0.953 for myopia onset) [91]. Another study by Li and colleagues

achieved strong predictive performance using multivariate linear regression for SE trajectory ($R^2 = 0.964$, MAE = 0.119 D) and logistic regression for predicting high myopia onset (AUC = 0.99) [53]. In addition, Lin *et al.* employed random forest models to predict SE progression and high myopia development over a 10-year horizon, reporting AUC values ranging from 0.862 to 0.958 [92]. These results collectively underscore the potential of ML techniques in forecasting RE progression with high accuracy (summarized in **Table 3**).

## Discussion

This study highlights the broad applications of AI technologies in the detection, diagnosis, prediction, progression, and treatment of REs. Overall, the findings suggest that AI holds significant potential to enhance the clinical management of REs in real-world settings, as demonstrated by consistently high sensitivity, specificity, and SROC values across various applications.

Our analysis demonstrated that AI, particularly, DL exhibited high accuracy in the detection and diagnosis of REs, with pooled SROC of 0.98, sensitivity of 95%, and specificity of 97%. This outstanding performance may be attributed to the established effectiveness of DL in ocular imaging tasks [94,95]. Deep learning models are especially advantageous due to their capacity to automatically extract and learn hierarchical and complex features from imaging modalities such as fundus photographs, thereby enhancing their utility in detection and diagnostic applications [96]. Another key finding from the meta-analysis of detection and diagnosis models was the significant impact of the input modality. Fundus images emerged as particularly effective, yielding an SROC greater than 0.98. This indicates that fundus photography offers reliable visual information essential for AI algorithms to accurately detect or diagnose REs, likely because it allows direct visualization of structural ocular changes associated with refractive conditions, particularly myopia [97]. Additionally, the robustness of DL performance may be partly attributed to the larger volume of studies employing this technique relative to other AI approaches, thereby offering more comprehensive evidence of its effectiveness.

In the prediction of REs, AI demonstrated high predictive performance, with a pooled SROC of 0.96 and pooled sensitivity and specificity of 87% and 96%, respectively. Our study found that DL models achieved a slightly higher sensitivity (0.89) compared to ML models (0.84). Conversely, ML models demonstrated higher specificity (0.99) compared to DL models (0.92), although these differences were not statistically significant. These results suggest that while DL models may be better at identifying positive cases, ML models may be more effective at reducing false positives, highlighting the importance of model selection based on specific clinical needs. It is important to note that the input data used by DL and ML models were not identical across the included studies. Notably, DL models in this category often utilized image-based modalities such as eccentric photorefraction images, which capture critical features relevant to refractive status, reinforcing the clinical utility of ocular appearance images in AI prediction. These findings indicate that AI, especially DL, could facilitate large-scale screening or early intervention, particularly in underserved or resource-limited settings where only ocular appearance images are available.

There has been growing interest in utilizing AI approaches to predict the progression of REs, particularly myopia and high myopia, over time. Nearly all the reviewed studies focusing on RE progression employed ML as their primary modeling technique. ML is well-suited for handling complex, multidimensional longitudinal patient data, especially in scenarios where clinical outcomes are not known in advance, enabling effective forecasting of disease trajectories [98,99]. We observed that all the included studies employed clinical features such as age, gender, parental myopia, and lifestyle factors like outdoor activity for predictive modeling—variables well-established as influencing myopia onset and progression [100,101]. Early identification of individuals exhibiting initial RE changes, particularly those at risk of developing high myopia, could facilitate timely therapeutic interventions aimed at slowing disease progression. This approach also supports personalized management strategies and better planning of future care. The AUC values reported across these progression studies ranged from 0.845 to 0.99, while $R^2$ values varied between 0.613 and 0.964, indicating that ML models have strong potential to accurately identify myopia progressors.

**Table 3. Characteristics of studies for predicting refractive error progression.**

| Author, Year | Data Used | Input features/ (No.) | AI Tech-nique | Best model | Target variable | Performance metrics |
|---|---|---|---|---|---|---|
| Varosanec *et al.* 2024 [52] | Clinical data | Total number = 15<br>Date of first and follow-up visits, school-age group, gender, age, correction method, uncorrected VA, best-corrected VA, best-corrected VA binocularly, baseline cycloplegic SE, corrected SE, sphere, cylinder, and axis, parental myopia status, myopia classification | DL | Extended gate time-aware long short-term memory | Future SE | MAE = 0.10 ± 0.15 D |
| Li *et al.* 2024a [53] | Clinical data | Total number = 4<br>Age at baseline, SE at baseline, The time interval between baseline and follow-ups, Corresponding outputs: SE at subsequent follow-up sessions | ML | Multivariate linear regression | SE progression trajectory | R² = 0.964<br>MAE = 0.119D |
| | | | ML | Logistic regression | High myopia onset | SEN = 98.96%<br>SPE = 93.70%<br>ACC = 94.31%<br>AUC = 0.99 |
| Barraza-Bernal *et al.* 2023 [84] | Clinical data | Total number = 3<br>SE, Age, AxL/Cr | ML | Support Vector Machines | Spherical power as a function of the age | R² = 0.57<br>RMSE = 1.33 D |
| Li *et al.* 2024b [87] | Clinical data | Total number = 1<br>SE | ML | XGBoost | Annual myopia progression | AUC = 0.96 |
| Zhao *et al.* 2024 b [91] | Clinical data | Total number = 3<br>Age at baseline, SE at the first examination, Annual myopia progression rate | ML | XGBoost | 1. SE over 15-year period<br>2. Onset of myopia<br>3. Onset of high myopia | R² = (0.613 -0.992)<br>MAE = (0.078 - 1.673)<br>MSE = (0.099 - 11.410)<br>SEN = (0.853 -0.967)<br>SPE = (0.53 - 0.986)<br>ACC= (0.854 -0.971)<br>AUC = (0.845 -0.953)<br>SEN = (0.682-1.00)<br>SPE = (0.804-0.994)<br>ACC= (0.784-0.994)<br>AUC = (0.765 - 0.997) |
| Lin *et al.* 2018 [92] | Clinical data | Total number = 3<br>Age at examination, SE, Annual progression rate | ML | Random forest | 1. SE in 10 years<br>2. Presence of high myopia in 10 years | AUC<br>3 years = 0.903 to 0.958<br>5years = 0.886 to 0.889<br>8 years = 0.862 to 0.888<br>MAE<br>3 years = 0.253 to 0.395<br>5 years = 0.394 to 0.496<br>8 years = 0.503 to 0.799 |

SE: Spherical Equivalent; ML: Machine Learning; AxL/Cr: Axial Length/Corneal Radius ratio; VA Visual acuity; UCVA uncorrected visual acuity; BCVA Best corrected visual acuity; MAE: Mean Absolute Error; RMSE: Root Mean Square Error; SEN: Sensitivity; SPE: Specificity; ACC: Accuracy; AUC: Area Under the Curve

As demonstrated in the results section, multiple studies have proposed various DL and ML models for the treatment of REs, particularly focusing on predicting Ortho-K lens parameters, selecting optimal laser refractive surgery procedures, evaluating myopia control outcomes, and guiding ICL interventions. However, there was notable inconsistency in how results were reported across these studies. While many studies reported metrics such as R², MAE, and accuracy, this was not standardized. The reported R² values ranged from 0.708 to 0.97, accuracies varied between 82.1% and 99.03%, and MAE values spanned from 0.054D to 6.008D. These

outcomes collectively underscore the precision and overall high performance of the AI models evaluated for treatment applications.

Of note, despite the comprehensiveness of our search strategy, we found no evidence of AI utilization in RE management across the African continent. This is particularly concerning given that Africa bears a substantial burden of UREs [102], and faces severe shortages of eye care professionals alongside significant challenges in delivering eye care services [103]. The marked scarcity of AI applications in RE management in this region highlights a significant healthcare disparity that urgently requires attention. In particular, this disparity can be attributed to multiple intersecting challenges, most notably the lack of high-quality, large-scale, and demographically diverse datasets from African populations. Most AI models in RE detection have been developed using datasets from high-income countries, often lacking representation of African ethnic and genetic diversity. As a result, such models may perform poorly when applied in African contexts, potentially leading to biased or inaccurate predictions [104–106]. Furthermore, clinical data collection systems in many African countries are either non-digitized or poorly maintained, with inadequate annotation, storage, and interoperability, limiting their usability for AI model training [104,107]. In addition to data limitations, there is a notable shortage of trained professionals in AI, data science, and medical image analysis across the continent [106,108]. Most healthcare workers and researchers have limited exposure to AI model development, validation, or implementation. Moreover, the few individuals who do receive AI training often face challenges such as inadequate infrastructure, limited access to GPUs, cloud computing services, and imaging technologies, further hindering innovation and local model development [107]. Beyond technical limitations, systemic barriers further complicate AI integration in African RE care. These include inadequate policy frameworks, limited funding for digital health innovation, ethical and privacy concerns, and low levels of community trust in AI technologies [109]. Language, cultural relevance, and the usability of AI tools also remain largely unexplored in current implementations, which are often designed for Western users and contexts. The adoption of advanced AI-based RE management systems could help bridge this gap by promoting equitable access to eye care services across Africa, thereby supporting the broader objectives of Sustainable Development Goal (SDG) 3 and the African Union's Agenda 2063 vision for transformed healthcare systems. However, successful implementation of AI in African settings will likely require careful and context-specific planning, taking into account factors such as specialist availability, long-term patient outcomes, and the cost-effectiveness of integrating AI into existing healthcare infrastructures, especially compared to resource-rich countries. Specifically, a multi-faceted strategy is needed. Investments should focus on establishing robust data infrastructure, building AI capacity through education and cross-disciplinary training, and creating inclusive datasets that reflect the diversity of African populations. Partnerships between local institutions, international collaborators, and technology developers can facilitate sustainable and ethical AI integration. Such initiatives will be critical to ensuring that AI tools are not only technically sound but are also socially acceptable and clinically relevant in this region.

## Limitations

Although this study presents the first comprehensive evidence synthesis highlighting the utility of AI in RE management, some limitations warrant consideration. First, by restricting our review to English-language publications, relevant studies in other languages may have been excluded. Notably, over half of the included studies were conducted in Asia, which may reflect regional differences in RE prevalence. However, future research involving more diverse geographic and ethnic populations is necessary to better assess the real-world performance and generalizability of AI across different settings. While all studies performed internal validation, the lack of external validation raises concerns about the broader applicability of these models, underscoring the need for further validation efforts. Most studies employed retrospective designs, which are susceptible to selection bias [110], and there remains a paucity of high-quality prospective studies evaluating AI performance in real-time clinical environments. Current guidelines emphasize that regulatory approval for AI devices requires validation through multicenter randomized controlled trials using standardized methods to ensure reliability and clinical

applicability [111]. Importantly, unlike RE detection, the long-term clinical benefits of AI, such as reducing the incidence and prevalence of REs, remain to be established, representing a critical outcome for future research [112].

## Conclusion

This systematic review and meta-analysis provide comprehensive evidence of the effectiveness and versatility of AI technologies, particularly DL and ML in the diagnosis, prediction, monitoring, and treatment of REs. Our findings indicate that AI models demonstrate high diagnostic accuracy, with pooled sensitivity and specificity exceeding 90%, particularly when using image-based inputs such as fundus photography. These results underscore the clinical potential of AI in improving early detection and intervention strategies for REs. In prediction and progression modeling, AI systems showed strong performance, with DL models favoring higher sensitivity and ML models offering better specificity. This suggests that model selection should be context-driven, depending on whether the goal is early case detection or minimizing false positives. Treatment-focused AI applications also showed promise, especially in supporting clinical decisions in orthokeratology, myopia control, and refractive surgeries, though variation in performance metrics and limited external validation call for cautious interpretation. A major strength of our review is its rigorous methodology, including adherence to PRISMA guidelines and the inclusion of a meta-analytic component. However, limitations such as reliance on English-language studies, retrospective designs, and a lack of studies from low-resource settings (particularly Africa) restrict generalizability. These gaps highlight significant disparities in AI adoption and the urgent need for inclusive datasets and equitable technological development. Future research should prioritize the development of interpretable, externally validated models trained on diverse, population-representative datasets. Strategic investments in data infrastructure, interdisciplinary collaboration, and digital capacity-building are critical, especially in underrepresented regions. Moreover, research should assess the real-world clinical impact, cost-effectiveness, and patient-centered outcomes of AI-assisted RE care. Taken together, AI holds transformative potential in RE management, and with responsible development and global inclusivity, it can serve as a scalable solution to address vision care challenges across diverse healthcare settings.

## Supporting information

**S1 PRISMA Check list. PRISMA ChecklistPRISMA (Preferred Reporting Items for Systematic Review and Meta-Analysis) checklist.**
(DOCX)

**S2 Text. The search strategy for each engine: a) PubMed, b) Web of Science, c) Embase, d) Scopus, e) Cochrane library and 5) Google scholar.**
(DOCX)

**S3 Table. Risk of bias of included studies using the Quality Assessment of Diagnostic Accuracy Studies-2 (QUADAS-2) tool.**
(XLSX)

**S4 Fig. Funnel plot asymmetry test for detection and diagnosis studies.**
(TIF)

**S5 Fig. Subgroup analysis plots by AI technique for detection and diagnosis studies.**
(TIF)

**S6 Fig. Subgroup analysis plots by input modalities for detection and diagnosis studies.**
(JPG)

**S7 Fig. Funnel plot asymmetry test for prediction studies.**
(TIF)

**S8 Fig. Subgroup analysis plots by AI technique for prediction studies.**
(JPG)

**S9 Fig. Subgroup analysis plots by input modalities for prediction studies.**
(JPG)

**S10 File. All studies identified in the literature search.**
(XLSX)

**S11 File. All data extracted from included studies.**
(XLSX)

**S12 File. Data used for meta-analysis.**
(XLSX)

## Author contributions

**Conceptualization:** Josephine Ampong, Sylvia Agyekum, Kwadwo Owusu Akuffo.

**Data curation:** Josephine Ampong, Albert Kwadjo Amoah Andoh, Clement Afari, Emmanuel Assan, Saphiel Osei Poku, Karen Ama Sam, Kwadwo Owusu Akuffo.

**Formal analysis:** Josephine Ampong, Sylvia Agyekum, Albert Kwadjo Amoah Andoh, Isaiah Osei Duah Junior, Kwadwo Owusu Akuffo.

**Investigation:** Josephine Ampong, Sylvia Agyekum, Werner Eisenbarth, Albert Kwadjo Amoah Andoh, Isaiah Osei Duah Junior, Gabriel Amankwah, Eldrick Adu Acquah, Saphiel Osei Poku, Karen Ama Sam, Josephine Ampomah Boateng, Kwadwo Owusu Akuffo.

**Methodology:** Josephine Ampong, Sylvia Agyekum, Werner Eisenbarth, Albert Kwadjo Amoah Andoh, Isaiah Osei Duah Junior, Gabriel Amankwah, Gabriel Kwaku Agbeshie, Eldrick Adu Acquah, Clement Afari, Emmanuel Assan, Saphiel Osei Poku, Karen Ama Sam, Josephine Ampomah Boateng, Kwadwo Owusu Akuffo.

**Project administration:** Josephine Ampong, Sylvia Agyekum, Werner Eisenbarth, Albert Kwadjo Amoah Andoh, Isaiah Osei Duah Junior, Gabriel Kwaku Agbeshie, Emmanuel Assan, Kwadwo Owusu Akuffo.

**Resources:** Josephine Ampong, Sylvia Agyekum, Werner Eisenbarth, Albert Kwadjo Amoah Andoh, Isaiah Osei Duah Junior, Gabriel Amankwah, Gabriel Kwaku Agbeshie, Clement Afari, Emmanuel Assan, Saphiel Osei Poku, Karen Ama Sam, Kwadwo Owusu Akuffo.

**Software:** Josephine Ampong, Sylvia Agyekum, Albert Kwadjo Amoah Andoh, Isaiah Osei Duah Junior, Gabriel Amankwah, Eldrick Adu Acquah, Clement Afari, Karen Ama Sam, Josephine Ampomah Boateng, Kwadwo Owusu Akuffo.

**Supervision:** Sylvia Agyekum, Kwadwo Owusu Akuffo.

**Validation:** Josephine Ampong, Sylvia Agyekum, Albert Kwadjo Amoah Andoh, Isaiah Osei Duah Junior, Clement Afari, Josephine Ampomah Boateng, Kwadwo Owusu Akuffo.

**Visualization:** Josephine Ampong, Sylvia Agyekum, Albert Kwadjo Amoah Andoh, Isaiah Osei Duah Junior, Kwadwo Owusu Akuffo.

**Writing – original draft:** Josephine Ampong, Sylvia Agyekum, Werner Eisenbarth, Albert Kwadjo Amoah Andoh, Isaiah Osei Duah Junior, Gabriel Amankwah, Kwadwo Owusu Akuffo.

**Writing – review & editing:** Josephine Ampong, Sylvia Agyekum, Werner Eisenbarth, Albert Kwadjo Amoah Andoh, Isaiah Osei Duah Junior, Gabriel Kwaku Agbeshie, Eldrick Adu Acquah, Clement Afari, Emmanuel Assan, Saphiel Osei Poku, Karen Ama Sam, Josephine Ampomah Boateng, Kwadwo Owusu Akuffo.

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
