## [Decision Letter · Decision Letter 0]

1 Aug 2025

Response to Reviewers
Revised Manuscript with Track Changes
Manuscript
**Journal Requirements:**

1. Please upload separate figure files in .tif or .eps format. Also, remove the figures from your manuscript file but keep the legends.

2. Please provide an Author Summary. This should appear in your manuscript between the Abstract (if applicable) and the Introduction, and should be 150–200 words long. The aim should be to make your findings accessible to a wide audience that includes both scientists and non-scientists. Sample summaries can be found on our website under Submission Guidelines: 

https://journals.plos.org/digitalhealth/s/submission-guidelines#loc-parts-of-a-submission

3. As required by our policy on Data Availability, please ensure your manuscript or supplementary information includes the following: 

4. We note that your Data Availability Statement is currently as follows: [Add Data Availability statement here]

**Additional Editor Comments (if provided):****Reviewers' Comments:** Reviewer #1: Line 43 of the abstract states that the work is “drawn from a pool of 8.086 studies”. However, this does not match the PRISMA diagram (Figure 1), which shows 6,288 articles were retrieved. Which is the correct number?

Similarly, also starting on line 43, “Nineteen of these studies were permissible for meta-analysis.” But the PRISMA diagram (Figure 1) shows that 45 studies were selected for inclusion. Which is the correct number? Or, is this mixing literature review with meta-analysis? If so, please make this clearer in the manuscript. Based on line 158, I suspect this is true.

The PRISMA diagram (Figure 1) is referred to in line 155 (results) but it would make more sense to be referred to in the method.

Referring to the above, what criteria enabled studies to be included in the review but prevented them from being included in the meta-analysis? This should be included in the method.

There are some minor grammatical errors; please carefully proofread the manuscript. Similarly, there are some minor typing errors; sometimes there is a space before the citation [1], and sometimes there is not[1].

Line 69 states, “Traditional methods for diagnosing and managing refractive errors rely on subjective assessments”. However, autorefraction tests are now fairly common. This doesn’t rely heavily on practitioner expertise or patient input. However. Autorefraction testing is not mentioned in this manuscript.

It may be worth visualising some of the study characteristics reported from line 166 and AI applications from line 176.

Some abbreviations, such as Machine Learning (ML), are defined multiple times.

In Table 1, I am not convinced that “Main Outcome” is the most appropriate title for that column. Perhaps something like target prediction, model aim, or something else.

In Tables 2 and 3, is “Best model” the correct column header? Some rows have more than one model. What dataset were these works evaluated on? Were they all evaluated on the same dataset and using the same methodology? If not, then the direct comparison in the performance metrics column may not make sense.

Reviewer #2: This paper presents an interesting systematic review of artificial

intelligence applications in th diagnosis and management of vision

problems associated with refractive errors. The paper is a

straightforward review with appropriate methods, a clear PRISMA

diagram ,and a readable and informative presentation of

results. Analyses by AI techniques, input modalities, and other

subgroups provide additional insight, along with descriptive

discussions of AI in treatment, and progression of refractive error.

The paper is interesting and informative.I have no concerns with the

analysis, or conclusions, but there are a few concerns with

some of the presentation, and some additional areas that would merit

from further elaboration:

1. The figures are of poor quality and should be recreated, or at the

very least re-submitted at better resolution. Figures 1 and 2 are

pixelated. Figures 3 and 4 and corresponding figures in the

supplemental document have a strange aspect ratio, as if they were

scaled unevenly in the vertical and horizontal. the SROC curves are

hard to read, consisting of what look to be a couple of small

squiggles.

2. The early presentation of the results (pages 15-18) is a bit

confusing to read, as sections are not as clearly delineated into

subsections as might be useful for making the paper more

readable.

3. There are two figures labeled "Figure 4".

4. Tables 2 and 3 provide a useful summary of the papers

discussed. However, potentially pertinent details are not

included. These tables would be more informative if expanded to

include a description of the sample sizes (number of patients/images),

and the number of features included.

5. The paper ends with an argument for the application of AI in

African settings. This is certainly an admirable goal worthy of

additional discussion. Additional elaboration regarding the factors

behind the limited discussion of AI in these settings would be

helpful. Specifically, are the limitations associated with a lack of

data? A shortage of skilled data scientists capable of creating the

appropriate AI models?

**Figure resubmission:****Reproducibility:** To enhance the reproducibility of your results, we recommend that authors of applicable studies deposit laboratory protocols in protocols.io, where a protocol can be assigned its own identifier (DOI) such that it can be cited independently in the future. Additionally, PLOS ONE offers an option to publish peer-reviewed clinical study protocols. Read more information on sharing protocols at https://plos.org/protocols?utm_medium=editorial-email&utm_source=authorletters&utm_campaign=protocols

---

## [Decision Letter · Decision Letter 1]

2 Sep 2025

Artificial intelligence applications in refractive error management: A systematic review and meta-analysis

PDIG-D-25-00300R1

Dear Dr. Akuffo,

We're pleased to inform you that your manuscript has been judged scientifically suitable for publication and will be formally accepted for publication once it meets all outstanding technical requirements.

Within one week, you'll receive an e-mail detailing the required amendments. When these have been addressed, you'll receive a formal acceptance letter and your manuscript will be scheduled for publication.

An invoice for payment will follow shortly after the formal acceptance. To ensure an efficient process, please log into Editorial Manager at https://www.editorialmanager.com/pdig/ click the 'Update My Information' link at the top of the page, and double check that your user information is up-to-date. For billing related questions, please contact billing support at https://plos.my.site.com/s/.

Kind regards,

Dhiya Al-Jumeily OBE, PhD

Section Editor

PLOS Digital Health

Reviewers' comments:

Reviewer #2: Thanks to the authors or this improved resubmission. My comments have

largely been addressed. However, I would like to request one minor

clarification:

In the abstract, the phrasing of the included papers is unclear: "A

total of 45 studies were included in the systematic review, drawn from

a pool of 6,288 studies. Nineteen of these studies were permissible

for meta-analysis. Of these, 55.5% (n= 25/45) utilized deep learning

(DL) and 44.4% (n =20/45) employed machine learning (ML) techniques."

This implies that 55.5% of the 19 studies in the meta-analysis used

deep learning, and 44.4% did not. I think the appropriate phrasing

would be as follows:

"A total of 45 studies were included in the systematic review, drawn from

a pool of 6,288 studies. Nineteen of these studies were permissible

for meta-analysis. Of the 45 included in the systematic review, 55.5%

(n= 25/45) utilized deep learning (DL) and 44.4% (n =20/45) employed

machine learning (ML) techniques."
